# The impact of vitamin D deficiency on some biochemical parameters and clinical outcome in Palestinian pregnant women during the first trimester

**Saleh Nazmy Mwafy**[1]*, **Safaa Ramadan Abed El- Nabi**[1], **Mohammed Marwan Laqqan**[2], **Maged Mohamed Yassin**[3]

1 Department of Biology, Faculty of Science, Al Azhar University-Gaza, Gaza Strip, Palestine, 2 Faculty of Health Sciences, Department of Laboratory Medical Sciences, Islamic University of Gaza, Gaza, Palestine, 3 Faculty of Medicine, Islamic University of Gaza, Gaza, Palestine

* smwafy@hotmail.com

**Data Availability Statement:** All relevant data are within the paper.

## Abstract

### Background and aim

Vitamin D deficiency is widespread worldwide and associated with negative effects on maternal and neonatal health. This study aimed to evaluate the relationship between vitamin D and thyroid and parathyroid hormone levels in the first trimester of pregnancy.

### Material and methods

This case control study included 200 participants aged (18–40) years divided into two groups; 100 pregnant females at the first trimester as case group, attending the main general clinical centers in Gaza strip, Palestine and 100 apparently healthy non-pregnant females as control group. Vitamin D, free thyroxine, free triiodothyronine, thyroid stimulating hormone, parathyroid, and the autoantibodies specific for thyroglobulin and thyroid peroxidase in serum were measured in all mothers and statistically analyzed using SPSS version 21 software.

### Results

Serum vitamin D, TSH, anti-TPO, and anti-TG levels were significantly decrease while, parathyroid levels were non-significantly decreased in the first trimester of pregnancy compered to control group. The levels of $fT_4$ were significantly increased and level of $fT_3$ were non significantly increased among pregnant mothers compered to control group. Analyses using Pearson correlation coefficients showed positive correlations between vitamin D with $fT_4$, $fT_3$, Anti-TPO with $P$-value < 0.05 and negative correlations with mother age, TSH, PTH with $P$-value < 0.05 in early pregnancy.

### Conclusions

Vitamin D deficiency among pregnant women in the first-trimester can be associated with thyroid, parathyroid parameters and thyroid autoantibodies with potential adverse

**Funding:** The authors received no specific funding for this work.

**Competing interests:** The authors have declared that no competing interests exist.

consequences for overall health, emphasizing a routine monitoring and vitamin D supplementation prevention strategies to optimize maternal and fetal outcomes.

## Introduction

Vitamin D is a fat-soluble steroid hormone and can be taken from meals or produced in the skin by sun exposure and UV light interaction with a cholesterol derivative [1]. The onset of vitamin D deficit or insufficiency in females of childbearing age is significantly influenced by poor dietary habits, sedentary lifestyles, and a lack of sun exposure [2]. Adequate vitamin D concentrations during pregnancy are necessary to ensure appropriate maternal responses to the calcium demands of fetus and neonatal handling of calcium. Vitamin D plays a vital role in calcium homeostasis and bone remodeling. As result, vitamin D deficiency during pregnancy leads to impaired bone development and fetal growth [3]. Pregnancy may be associated with changes in iodine homeostasis and other physiological changes that eventually lead to altered thyroid function, consequently normal mother thyroid function in the first trimester should be maintained, because the growth of fetus brain depends completely on maternal thyroid hormones [4]. Maternal thyroid dysfunction is related to an increased risk of different adverse maternal and child outcomes such as miscarriage, intrauterine growth retardation, hypertensive disorders and preterm delivery [5]. Pregnant females with positivity thyroid autoantibody are correlated with maternal and fetal outcomes such as subfertility, recurrent embryo implantation failure, miscarriage, gestational diabetes, placental abruption, preterm rupture of membranes and hydramnios [6]. Since thyroid hormones and vitamin D receptors are similar to steroid hormone receptors, any alteration in a woman's constructor genes increases their risk of developing autoimmune disorders, such as thyroid autoimmune diseases (such as Hashimoto and Graves). The risk of thyroid problems may therefore increase with any change in vitamin D levels. For these reasons, it's crucial for sufferers with thyroid disease to realize how vitamin D affects their health [7]. Parathyroid is the main factor of bone and mineral metabolism through maintain calcium and phosphate homeostasis. In pregnancy, the calcium demand of the fetus results in profound changes in maternal calcium homeostasis, whereas parathyroid plays a major role in calcium and bone metabolism in the non-pregnant state, vitamin D appears to be a prominent regulator during pregnancy [8]. Several studies have shown that vitamin D deficiency and thyroid function can lead to a series of adverse health disorders during pregnancy, including gestational hypertension, cardiovascular disease, metabolic health, diabetes, multiple sclerosis, bone strength and cancers [9, 10]. This study was designed to explore the relationship between vitamin D and thyroid and parathyroid hormone levels in Palestinian pregnant females during the first trimester of pregnancy.

## Materials and methods

### Study population and experimental design

This case control study comprised 200 females aged 18–40 years in Gaza strip and divided into two groups. The first group included 100 pregnant females at the first trimester (gestational weeks 6–12) as case group. The second group included 100 apparently healthy and non-pregnant females as control group for baseline comparisons. The study was conducted at the main general clinical centers in Gaza city and North Gaza: Al- Remal Clinical Center, Al- Sorani Clinical Center and Jabaliya Clinical Center in Gaza strip during the period from April, 2019 to April, 2020. The sample size calculations were based on the formula of EPI-INFO statistical

package version 3.5.1 was used with 95% CI, 80% power and 50% proportion as conservative and OR>2. The sample size in case of 1:1 ratio of case control was 93. For a no-response expectation, the sample size was increased to 100 females and the control also consisted of 100 healthy females.

**Inclusion criteria.** In the study were: age 18–40 years, gestational age from 6–12 weeks, not have a chronic disease, and malabsorption, single pregnancy, not taking any supplements except multivitamins, and Palestinian nationality. Controls and cases were matched by frequency matching method in terms of age, number of pregnancies, age of marriage.

**Exclusion criteria.** Were mothers who under 18 years old and over 40 years old, mothers who refused to continue in the study, mothers who followed a special diet or were vegetarians, and mothers who had hypothyroidism prior to becoming pregnant and used levothyroxine tablets, heparin, glucocorticoids, or beta-blockers, hormone therapy, and females with chronic illnesses or those receiving special medical treatments during pregnancy were excluded from the study.

**Ethics.** For ethical consideration, the necessary approval to conduct this study was obtained from Helsinki committee with ethical approval number PHRC/HC/559/19 in the Gaza Strip and approval letter signed from the Palestinian Ministry of Health (MOH). All participants freely signed the written informed consent form of the study and freely participated in the study.

**Body mass index measurements.** BMI measurements were taken for each participant in light clothing and without shoes. Height and weight were measured by automatic height-weight scale (Detecto DR400C, USA), to the nearest 0.1 cm and 0.1 kg, respectively. Weight (kg) divided by height ($cm^2$) was used to compute BMI ($m^2$).

**Specimen collection and processing.** Five ml of venous blood was collected from each participant at the morning after 12 hours overnight fast in a lavender top tube, allowed to stand at room temperature for 20 minutes before being centrifuged for 10 minutes at 3000 rpm to separate the serum for biochemical analysis.

**Biochemical analysis.** Vitamin D was quantitatively determined by a microplate enzyme immunoassay using AccBind ELISA microwells kit (Monoband Inc., lake forest, USA) [11]. Free thyroxine, free triiodothyronine, thyroid stimulating hormone and parathyroid concentrations were measured by using ELISA methods [12]. The autoantibodies specific for thyroglobulin and thyroid peroxidase in serum were quantitatively determine by a microplate enzyme immunoassay [13].

**Data analysis.** The IBM SPSS program for Windows, version 21.0, was used to tabulate, and statistically analyze the data. Data analysis was carried out as follows: data cleaning, frequency table for all the study variables. Frequency and proportions were used to express qualitative data, defining and recording of certain variables and cross tabulation and advanced statistical analysis. The following tests were applied; ANOVA test, the significance of difference was checked by one-way ANOVA test. Chi-square test was used to determine how the proportions differed. Spearman correlation coefficient and independent-samples t-test were also used to calculate the variance in means across groups. P-values less than 0.05 were regarded as the statistically significant limit. Percentage change was calculated according to the following formula:

$$\textbf{Change percentage } (\%) = \frac{\textbf{mean of cases} - \textbf{mean of control}}{\textbf{mean of control}} \textbf{x100}$$

**Table 1. Basic characteristics of participants in case and control groups.**

| Character | Category | Vitamin D status | | | | χ2 | P-value |
|---|---|---|---|---|---|---|---|
| | | Sufficient group | | Deficient group | | | |
| | | No. | % | No. | % | | |
| Age (years) | 18–25 | 23.0 | 23.0 | 57.0 | 57.0 | 36.7 | 0.001 |
| | 26–33 | 45.0 | 45.0 | 39.0 | 39.0 | | |
| | 34–40 | 32.0 | 32.0 | 4.0 | 4.0 | | |
| | Mean ± SD | 25.4±5.5 | | 25.0±4.4 | | | |
| Weeks of gestation | 5–8 | 0.0 | 0.0 | 49.0 | 49.0 | 200.0 | 0.001 |
| | 9–12 | 0.0 | 0.0 | 51.0 | 51.0 | | |
| Educational status | Primary | 18.0 | 18.0 | 8.0 | 8.0 | 7.6 | 0.023 |
| | Secondary | 38.0 | 38.0 | 55.0 | 55.0 | | |
| | University | 44.0 | 44.0 | 37.0 | 37.0 | | |
| Employment status | Yes | 17.0 | 17.0 | 6.0 | 6.0 | 6.0 | 0.015 |
| | No | 83.0 | 83.0 | 94.0 | 94.0 | | |

No.: number of the subject; SD: standard deviation; $\chi^2$: chi-square test.

Each reading represents Mean ±SD of 100 subjects.

*The significant of difference was checked by chi square test, significant at $P < 0.05$.

## Results

A total of 200 females were recruited in this study as two groups; the case (with vitamin D deficiency) include 100 females were in the first trimester of pregnancy aged from 18 to 40 years with mean age at blood draw was 25.0±4.4 years and the control group (without vitamin D deficiency) include 100 non-pregnant females with mean age 25.4±5.5 years ($\chi^2 = 36.7$, $P = 0.001$). The basic characteristics of the females are summarized in "Table 1".

The greater percentage of pregnant mothers (57.0%) were in the younger age group (18–25 years), whereas 43.0% of pregnant mothers in the age group 26–33 years and 34–40 years. Among the participants in the first trimester of pregnancy 49.0% of cases were at 5–8 weeks of pregnancy and 51.0% were at 9–12 week of gestation. Analysis of educational status showed that 8.0% of cases and 18.0% of controls were had primary school, while 55.0% of cases and 38.0% of controls had finished secondary school and 37.0% of cases and 44.0% of controls had finished university with ($\chi^2 = 7.6$, $P = 0.023$). More than 80.0% of cases and of controls were unemployed and less than 20.0% of the study participants were employed. As indicated in "Table 2", the BMI was significantly increased in pregnant mothers compared to non-pregnant control group with 15.6% percent of change ($P < 0.001$).

Table 3. showed that in the first trimester, the mean serum vitamin D were significantly decreased in pregnant mothers compered to non-pregnant control group (14.7±8.8 vs 37.2

**Table 2. Anthropometric measurements of participants in case and control groups.**

| Anthropometric measurement | Sufficient group | Deficient group | % Change | P-value |
|---|---|---|---|---|
| **Height** (cm) | 159.1±5.8 | 159.9±5.4 | 0.5 | 0.291 |
| **Weight** (kg) | 64.9±13.1 | 75.6±13.2 | 16.5 | 0.001 |
| **BMI** (Kg/m$^2$) | 25.6±4.8 | 29.6±5.3 | 15.6 | 0.001 |

Kg: kilogram, cm: centimeter, BMI: Body mass index: People with BMI = 18.5.24.9 were considered to have normal weight and people with BMI 25–29.9 were classified overweight.

Each reading represents Mean ±SD of 100 subjects. The significance of difference was checked by t-test, significant at $P \leq 0.05$.

**Table 3. Vitamin D status of participants in case and control groups.**

| Parameter | N (%) | Sufficient group | Deficient group | % Change | P-value |
|---|---|---|---|---|---|
| **Vitamin D levels** (ng/ml) | 100 | 37.2±4.8 | 14.7±8.8 | -60.5 | 0.001 |
| **Vitamin D status** | | | | | |
| Severe deficiency | 16.0 | - | 9.4±0.4 | -74.8 | 0.001 |
| Deficiency | 60.0 | - | 11.3±0.9 | -69.6 | |
| Insufficiency | 24.0 | - | 24.2±1.5 | -34.8 | |

Vitamin D status based on serum vitamin D levels: severe vitamin D deficiency, serum 25OHD level<5-≤10 ng/mL; vitamin D deficiency, serum 25OHD levels >10-≤20 ng/mL; insufficient vitamin D, serum 25OHD levels >20-≤30 ng/mL; and sufficient vitamin D, serum 25OHD levels >30 ng/mL

Each reading represents Mean ±SD of subjects.

The significant of difference was checked by one-way ANOVA test (compare all vs. control), significant at $P \leq 0.05$.

±4.8 ng/mL) with percent of change 60.5% and $P< 0.001$. Severe vitamin D deficiency was observed in 16.0% of case (9.4±0.4 ng/mL), deficient vitamin D in 60.0% of pregnant females (11.3±0.9 ng/mL), and insufficient vitamin D in 24.0% of participants (24.2±1.5 ng/mL).

Table 4 shows that TSH levels (1.8±1.2 vs. 2.3±1.1 μIU/ml) were significantly lowered in cases compered to control group with percentage change of -21.7 and $P< 0.003$. Conversely, the mean levels of $fT_4$ (1.4±0.3 vs 1.2±0.3 ng/ml) were significantly higher in pregnant mothers compared to nonpregnant control group with percentage change of 13.9 and $P< 0.001$. Non significant increase in $fT_3$ mean levels among pregnant mothers compered to control ($P = 0.300$). The mean values of thyroid autoantibodies of the different study groups revealed that the mean levels of anti-TPO were 27.9±9.7 vs 33.1±11.9 IU/ml and anti-TG (39.3±18.5 vs 51.3±21.9 IU/ml) were significantly lower among pregnant females in the first trimester compared to controls $P< 0.001$. Parathyroid levels were non-significantly decreased in cases compered to control group (24.7±27.7 vs 26.8 ±18.2 pg/ml) with percent change of -7.8 and $P = 0.540$.

Analyses using Pearson correlation and linear regression coefficient was performed to assess the relationship between vitamin D and the study parameters and to evaluate the effect of vitamin D deficiency with some potentially effective parameters including mother age, gestational weeks, BMI, thyroid function tests (free T3, free T4, and TSH levels), thyroid autoantibodies (Anti-TPO, Anti-TG) and parathyroid hormones "Table 5". Among these correlations, significant positive correlations were found between vitamin D with $fT_4$ (r = 0.243), $fT_3$ (r = 0.272), Anti-TPO (r = 0.297) with $P$-value < 0.05 in the first trimester of pregnancy.

**Table 4. Thyroid autoantibody, thyroid and parathyroid hormones levels in study population.**

| Parameter | Sufficient group | Deficient group | % Change | P-value |
|---|---|---|---|---|
| **TSH** (μIU/ml) | 2.3±1.1 | 1.8±1.2 | -21.7 | 0.003 |
| **$fT_4$** (ng/dl) | 1.2±0.3 | 1.4±0.3 | 13.9 | 0.001 |
| **$fT_3$** (pg/ml) | 2.40±0.70 | 2.50±0.6 | 4.2 | 0.300 |
| **Anti-TPO** (IU/ml) | 33.1±11.9 | 27.9±9.7 | -15.7 | 0.001 |
| **Anti-TG** (IU/ml) | 51.3±21.9 | 39.3±18.5 | -23.5 | 0.001 |
| **PTH** (pg/ml) | 26.8±18.2 | 24.7±27.7 | -7.8 | 0.540 |

TSH: Thyroid stimulating hormone, $T_3$: Triiodothyronine hormone, $T_4$: Thyroxin hormone, Anti-TPO: anti-thyroid peroxidase, Anti-TG: anti-thyroglobulin, PTH: parathyroid hormone.

Each reading represents Mean ±SD of 100 subjects.

The significant of difference was checked by t-test, significant at $P \leq 0.05$.

**Table 5. Linear correlations between vitamin D and study parameters.**

| Parameters | Vitamin D | | |
|---|---|---|---|
| | r | βª | *P*-value |
| **Age of Mother** (year) | -0.195** | -0.204* | <0.05 |
| **Weeks of gestation** | 0.176 | 0.138 | >0.05 |
| **BMI(Kg/m2)** | 0.063 | 0.109 | >0.05 |
| **TSH** (μIU/ml) | -0.620* | -0.530* | <0.05 |
| **fT₄** (ng/dl) | 0.243** | 0.150* | <0.05 |
| **fT₃** (pg/ml) | 0.272** | 0.270** | <0.05 |
| **Anti-TPO** (IU/ml) | 0.297** | 0.180** | <0.05 |
| **Anti-TG** (IU/ml) | 0.104 | 0.032 | >0.05 |
| **PTH** (pg/ml) | -0.164* | -0.181** | <0.05 |

r: Pearson correlation, βª: Beta coefficient

* Correlation is significant at the 0.05 level (2-tailed).

**Correlation is significant at the 0.01 level (2-tailed). FT4: free thyroxin, FT3: free tri-iodothyronine, Anti-TPO: anti- thyroid peroxidase, Anti-TG: anti-thyroglobulin, PTH: parathyroid hormone.

Statistically significant negative correlations were found between vitamin D with mother age (r = -0.195, β: -0.204), TSH (r = -0.620, β: -0.530), PTH (r = -164, β: -0.181) with *P*-value < 0.05 in early pregnancy.

## Discussion

This study aimed to investigate the relations between vitamin D deficiency with thyroid and parathyroid hormone in the first trimester of pregnancy at the main general clinical centers in Gaza city and North Gaza, Palestine, during April 2019 –April 2020. The results revealed that the mean levels of BMI were significantly higher in pregnant females compared to control. This result was in agreement with [14]. The weight gain during pregnancy may be due to fetus, amniotic fluid, placenta, uterine and breast hypertrophy. The placenta of the fetus and amniotic fluid account for about 35% of total weight gain during pregnancy [15]. During gestation, lipid metabolism changes because of an increase in hepatic lipase activity, a decrease in lipoprotein lipase activity, delayed uptake of the remnant chylomicrons and hormonal changes [16]. The result of this study indicated that vitamin D deficient females in the first trimester showed significant lower vitamin D levels compared to controls. In addition, 16% of studied pregnant females had severe vitamin D deficiency, 60% had vitamin D deficiency and 24% had insufficient vitamin D. Similar data have been reported from other studies [17, 18]. The cut-off values for vitamin D status were recommended by endocrine society clinical practice guideline; females who had serum 25(OH) D levels less than 10 ng/mL were in the severe deficiency, those who had levels <20 ng/ml was defined as vitamin D deficiency, insufficiency as levels <30 ng/ml and sufficiency as levels of ≥30 ng/ml [19]. The minimum consensus in the scientific community is that serum 25(OH) D levels <20 ng/mL must be prevented and treated. Vitamin D deficiency identified as a pandemic, and it has a significant impact in health and preventing disease [20, 21]. The pregnant woman is the only source of vitamin D for the fetus, therefore, sufficient amount of vitamin D during pregnancy is essential for the growth and development of the fetus as it raises the amount of calcium needed to produce fetal bone growth [22]. Pregnant females should satisfy vitamin D needs of both mother and fetus vitamin D through sunlight, dairy products, oily salmon, and dietary supplements. Several recent studies have documented the association between pregnant vitamin D deficiency and adverse

outcomes in the mother including preeclampsia, pregnancy induced hypertension, gestational diabetes, obstructed labor, vaginosis, and cesarean delivery, additionally, to neonatal hypoglycemia, hyperinsulinemia, decreased birth height, weight, and head circumference of the child [23, 24]. Result of current study showed that $fT_4$ was higher in the pregnant, while the TSH was lowered with significant different at $P<0.05$. The levels of fT3 were non significantly increased in pregnant females. The study's findings are consistent with that obtained recently [25]. The decrease in serum TSH observed during the first trimester may be due to elevated levels of serum human gonadotropic hormone (HCG) that stimulating the TSH receptor and thereby increasing thyroid hormones production [26]. HCG is produced from the 8[th] day of pregnancy, initially by the embryo through the syncytiotrophoblast. Due to the structural similarity to TSH, it acts as a stimulating agent for the thyroid. The HCG hormone has a weak thyrotropic activity is only about 1/10 as potent as TSH during normal pregnancy through the activation of the TSH receptor. HCG action reaches its peak at the end of the first trimester (max. at 9[th] to 11[th] week of gestation) and then gradually decreases [27]. During pregnancy, there is an increase in T4 production in response to the estrogen hormone that increase thyroid hormone transport protein concentration, especially thyroxine binding globulin (TBG). The consequent fetal consumption of maternal thyroid hormones, together with high concentration of TBG, elevating urinary iodide clearance and increasing thyroid hormones degradation by placental D3, necessitates an increase in maternal thyroid hormones production to ensure adequate fetal thyroid hormones availability. Moreover, HCG has a stimulatory impact on thyroid hormone synthesis and release by binding to the TSH receptor on thyrocytes [28]. Serum $fT_4$ measurements in pregnant females are complicated by increased TBG and decreased albumin concentrations. High TBG concentrations in serum samples tend to result in higher $fT_4$ values, whereas low albumin in serum likely will yield lower $fT_4$ values.

Greater fT4 values are often associated with higher TBG concentrations in blood samples, whereas lower fT4 values are likely to be associated with lower albumin concentrations in serum [29]. Our findings revealed significant positive correlations between vitamin D with $fT_4$, $fT_3$, and negative correlation with TSH during early pregnancy. It seems that vitamin D deficiency in the first trimester of pregnancy may increase $fT_4$ and $fT_3$ levels and decrease TSH. Iodine homeostasis alterations and other physiological changes that may be brought on by pregnancy might in the end result in changes to thyroid function [30]. The maternal thyroid function should be kept normal during the first trimester, because the fetus's complete dependence on the mother's thyroid hormones throughout the first trimester of pregnancy. Therefore, an optimal vitamin D level should be maintained to ensure fetal health, a healthy pregnancy, appropriate fetal skeletal development and to prevent preeclampsia. The mean levels of anti-TPO and anti-TG were significantly lowered among pregnant group compared to control group. Our results were matched with others [31]. The decline in the level of thyroid autoantibodies (anti-TPO and anti-TG) may be due to an immunosuppressive effect of pregnancy on the thyroid function [31]. Thyroperoxidase (TPO) is a membrane-bound enzyme, which catalyzes iodide oxidation and iodination of tyrosyl residues of thyroglobulin (TG). Anti-TPO antibody can reacts with TPO leading to the destruction of thyrocytes. Autoantibodies to TPO are common in the euthyroid population and are associated with major alterations in the course of pregnancy affecting the mother, fetus and/or neonate. Anti-TPO is associated with a higher risk of pregnancy complications such as placental abruption, miscarriage preterm delivery and pregnancy-induced hypertension [32]. Our findings showed that the mean levels of parathyroid were not significantly decreased and there were significant negative correlations between vitamin D and parathyroid in the first trimester of pregnancy. This result was matched with previous study [33]. Vitamin D and parathyroid hormone (calciotropic hormones) control serum calcium and maintain whole-body calcium homeostasis.

Parathyroid hormone is calcium homeostatic hormones, is essential to increase maternal calcium absorption during pregnancy [34]. PTH has a role in a number of functions, including the maintaining the ionized calcium in blood, elevating the level of calcium-phosphate released from bone tissue, calcium conserving, lowering tubular phosphate reabsorption, and enhancing intestinal calcium absorption via vitamin D. PTH is lowered to the lower end or just below the optimal reference range in the first trimester of pregnancy [35]. Parathyroid is affected by maternal dietary calcium intake as well as maternal vitamin D levels and tightly regulates renal production of 25(OH) D as well as serum calcium and phosphorus levels. Therefore, there is an inverse relationship between serum 25(OH) D and parathyroid hormone.

## Conclusions

Vitamin D deficiency among pregnant women in the first-trimester can be associated with thyroid, parathyroid parameters and thyroid autoantibodies with potential adverse consequences for overall health, emphasizing a routine monitoring and vitamin D supplementation prevention strategies to optimize maternal and fetal outcomes.

## Author Contributions

**Conceptualization:** Saleh Nazmy Mwafy, Maged Mohamed Yassin.

**Formal analysis:** Saleh Nazmy Mwafy, Safaa Ramadan Abed El- Nabi.

**Investigation:** Saleh Nazmy Mwafy, Safaa Ramadan Abed El- Nabi, Mohammed Marwan Laqqan.

**Methodology:** Saleh Nazmy Mwafy, Safaa Ramadan Abed El- Nabi, Mohammed Marwan Laqqan.

**Project administration:** Saleh Nazmy Mwafy.

**Resources:** Maged Mohamed Yassin.

**Supervision:** Saleh Nazmy Mwafy.

**Validation:** Saleh Nazmy Mwafy.

**Visualization:** Saleh Nazmy Mwafy.

**Writing – original draft:** Saleh Nazmy Mwafy.

**Writing – review & editing:** Saleh Nazmy Mwafy, Mohammed Marwan Laqqan, Maged Mohamed Yassin.

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
