## [Decision Letter · Decision Letter 0]

23 Jan 2023

PONE-D-23-00231The Impact of Vitamin D Deficiency on Some Biochemical Parameters and Clinical Outcome in Palestinian Pregnant Women During the First TrimesterPLOS ONE

Dear Dr. Mwafy,

Thank you for submitting your manuscript to PLOS ONE. After careful consideration, we feel that it has merit but does not fully meet PLOS ONE’s publication criteria as it currently stands. Therefore, we invite you to submit a revised version of the manuscript that addresses the points raised during the review process.

We look forward to receiving your revised manuscript.

Kind regards,

Kornelia Zaręba, MD

Academic Editor

PLOS ONE

Journal Requirements:

- 10.4103/0976-9668.160021 

- https://doi.org/10.1089/thy.2011.0087

- https://doi.org/10.1007/s00404-018-5018-8

In your revision ensure you cite all your sources (including your own works), and quote or rephrase any duplicated text outside the methods section. Further consideration is dependent on these concerns being addressed.

Reviewers' comments:

Reviewer's Responses to Questions

**Comments to the Author**

1. Is the manuscript technically sound, and do the data support the conclusions?

Reviewer #1: Yes

Reviewer #2: Yes

2. Has the statistical analysis been performed appropriately and rigorously? 

Reviewer #1: Yes

Reviewer #2: Yes

3. Have the authors made all data underlying the findings in their manuscript fully available?

Reviewer #1: Yes

Reviewer #2: Yes

4. Is the manuscript presented in an intelligible fashion and written in standard English?

Reviewer #1: Yes

Reviewer #2: Yes

5. Review Comments to the Author

Reviewer #1: Thyroid health during pregnancy is very important both mother and fetus. Although n number is low it is found relationship between Vitamine D levels and TSH and FT4. This study has relevant results for clinical purposes.

Reviewer #2: The subject is relevant and deserves to be encouraged. The point is the form of your tables. The tables should be scientific.

Did you follow ethical procedures?

Thank you for trusting us to review the articles

6. PLOS authors have the option to publish the peer review history of their article (what does this mean?). If published, this will include your full peer review and any attached files.

Reviewer #1: No

Reviewer #2: **Yes: **Professeur Emérite Dr LONGO-MBENZA Benjamin

---

## [Author Response · Author response to Decision Letter 0]

12 Feb 2023

Suggestions

Corrections

We would like to thanks supporting us with valuable comments, we address these additional requirements according to the PLOS ONE style templates in the provided links and highlighting in red color.

Suggestions

Corrections

We respect your comments and specified in the ethics statement in the Methods that we obtained written informed consent form. 

For ethical consideration, the necessary approval to conduct this study was obtained from Helsinki committee with ethical approval number PHRC/HC/559/19 in the Gaza Strip and approval letter signed from the Palestinian Ministry of Health (MOH). All participants freely signed the written informed consent form of the study and freely participated in the study.

Suggestions

- 10.4103/0976-9668.160021 

- https://doi.org/10.1089/thy.2011.0087

- https://doi.org/10.1007/s00404-018-5018-8

In your revision ensure you cite all your sources (including your own works), and quote or rephrase any duplicated text outside the methods section. Further consideration is dependent on these concerns being addressed.

Corrections

We cite all of the sources including our works, and rephrase any duplicated text.

- HCG has a stimulatory impact on thyroid hormone synthesis and release by binding to the TSH receptor on thyrocytes.

- Greater fT4 values are often associated with higher TBG concentrations in blood samples, whereas lower fT4 values are likely to be associated with lower albumin concentrations in serum.

- Anti-TPO is associated with a higher risk of pregnancy complications such as placental abruption, miscarriage preterm delivery and pregnancy-induced hypertension

Suggestions

Corrections

We add in the title page the following data availability statement: All relevant data are within the manuscript and fully available without restriction.

Reviewer Suggestions

1. Is the manuscript technically sound, and do the data support the conclusions?

Reviewer #1: Yes

Reviewer #2: Yes

Corrections

We thank both of the reviewers for their kind comment and we critically read and revised our manuscript and highlighting the changes in red color.

Reviewer Suggestions

2. Has the statistical analysis been performed appropriately and rigorously?

Reviewer #1: Yes

Reviewer #2: Yes

Corrections

We made the following tests were applied; ANOVA test, the significance of difference was checked by one-way ANOVA test. Chi-square test was used to determine how the proportions differed. Spearman correlation coefficient and independent-samples t-test were also used to calculate the variance in means across groups. 

Reviewer Suggestions

3. Have the authors made all data underlying the findings in their manuscript fully available?

Reviewer #1: Yes

Reviewer #2: Yes

Corrections

We add in the title page the following data availability statement: All relevant data are within the manuscript and fully available without restriction.

Reviewer Suggestions

4. Is the manuscript presented in an intelligible fashion and written in standard English?

Reviewer #1: Yes

Reviewer #2: Yes

Corrections

Our collogue critically read our manuscript and revised the English and typo as required.

Reviewer Suggestions

5. Review Comments to the Author

Reviewer #1: Thyroid health during pregnancy is very important both mother and fetus. Although n number is low it is found relationship between Vitamine D levels and TSH and FT4. This study has relevant results for clinical purposes.

Reviewer #2: The subject is relevant and deserves to be encouraged. The point is the form of your tables. The tables should be scientific.

Did you follow ethical procedures?

Thank you for trusting us to review the articles

Corrections

We would like to thanks Reviewer #1 for spending their valuable time in revision our manuscript and supporting us with valuable comments. Thank for your kind word that describing the important and clinical utility of vitamin D deficiency on some biochemical parameters and clinical outcome in Palestinian pregnant women during the first trimester and the important of thyroid for mother and child. 

We respect your decision concerning our manuscript. However, we critically re-read the manuscript and re-revised the English and typo and we re-standardized and re-format all tables in our manuscript

We would like to thanks Reviewer #2 for spending their valuable time in revision our manuscript and supporting us with valuable comments. We respect your decision concerning our manuscript and followed your advice and we critically re-read the manuscript and re-revised the English and typo and re-standardized and re-format all tables in our manuscript.

We respect your comments and specified in the ethics statement in the Methods that we obtained written informed consent form. 

For ethical consideration, the necessary approval to conduct this study was obtained from Helsinki committee with ethical approval number PHRC/HC/559/19 in the Gaza Strip and approval letter signed from the Palestinian Ministry of Health (MOH). All participants freely signed the written informed consent form of the study and freely participated in the study.

---

## [Editor Report · Decision Letter 1]

8 Mar 2023

The Impact of Vitamin D Deficiency on Some Biochemical Parameters and Clinical Outcome in Palestinian Pregnant Women During the First Trimester

PONE-D-23-00231R1

Dear Dr. Saleh Nazmy Mwafy

We’re pleased to inform you that your manuscript has been judged scientifically suitable for publication and will be formally accepted for publication once it meets all outstanding technical requirements.

Kind regards,

Kornelia Zaręba, MD

Academic Editor

PLOS ONE

---

## [Editor Report · Acceptance letter]

20 Mar 2023

PONE-D-23-00231R1 

The impact of vitamin D deficiency on some biochemical parameters and clinical outcome in Palestinian pregnant women during the first trimester 

Dear Dr. Mwafy:

I'm pleased to inform you that your manuscript has been deemed suitable for publication in PLOS ONE. Congratulations! Your manuscript is now with our production department. 

Kind regards, 

on behalf of

Dr. Kornelia Zaręba 

Academic Editor

PLOS ONE